# Single-Round LDCT Screening in Men Aged ≥ 70 Years: Prevalence of Pulmonary Nodules and Lung Cancer Detection

**DOI:** 10.3390/cancers17142318

**Published:** 2025-07-11

**Authors:** Hye-Rin Kang, Jin Hwa Song, Yeon Wook Kim, Keun Bum Chung, Sukki Cho, Seung Hun Jang, Jin-Haeng Chung, Jaeho Lee, Choon-Taek Lee

**Affiliations:** 1Division of Pulmonology and Allergy, Department of Internal Medicine, Hallym University Dongtan Sacred Heart Hospital, Hwaseong 18450, Republic of Korea; kang@hallym.or.kr (H.-R.K.);; 2Division of Pulmonary and Critical Care Medicine, Department of Internal Medicine, Seoul National University College of Medicine, Seoul 03080, Republic of Koreajhlee7@snubh.org (J.L.); 3Division of Pulmonary and Critical Care Medicine, Department of Internal Medicine, Seoul National University Bundang Hospital, Seongnam 13620, Republic of Korea; 4Division of Pulmonary, Allergy, and Critical Care Medicine, Department of Internal Medicine, Veterans Health Service Medical Center, Seoul 05368, Republic of Korea; 5Department of Thoracic and Cardiovascular Surgery, Seoul National University Bundang Hospital, Seongnam 13620, Republic of Korea; 6Department of Pulmonary, Allergy and Critical Care Medicine, Hallym University Sacred Heart Hospital, Anyang 14068, Republic of Korea; 7Department of Pathology and Translational Medicine, Seoul National University Bundang Hospital, Seongnam 13620, Republic of Korea

**Keywords:** lung cancer screening, lung neoplasms, aged, pulmonary nodules, low-dose computed tomography

## Abstract

Lung cancer screening with low-dose computed tomography (LDCT) has proven effective, yet its application in older adults (aged 70+) remains underexplored. This study investigated a single round of LDCT screening in 1409 elderly Korean men (average age 74.2 years) to assess pulmonary nodule prevalence and lung cancer detection. We found a high prevalence of pulmonary nodules (55.8%), with 12.7% classified as positive. Crucially, a lung cancer detection rate of 2.2% was observed, which is higher than rates reported in major trials. Over half of these cancers were diagnosed at an early stage (I or II). While these findings suggest a potential benefit of early detection in this older, predominantly male cohort, the retrospective nature and inherent population bias necessitate further prospective studies to confirm these promising results and inform broader screening guidelines.

## 1. Introduction

Lung cancer remains the leading cause of cancer-related death worldwide [1]. Low-dose computed tomography (LDCT) screening has proven effective in reducing mortality through early detection, as demonstrated by major randomized controlled trials (RCTs) such as the National Lung Screening Trial (NLST) and Nederlands–Leuvens Longkanker Screenings ONderzoek (NELSON) [2,3]. These landmark studies primarily targeted high-risk smokers who were aged 50–74 years, showing a 20–24% reduction in lung cancer mortality [2,3]. Other European trials—including DANTE, ITALUNG, MILD, LUSI, and UKLS—have further explored LDCT protocols but maintained similar age restrictions (≤80 years) [4,5,6,7,8]. More recently, lung cancer screening studies have begun to include never-smokers in their populations. The largest RCT in this population, the Taiwan Lung Cancer Screening in Never-Smoker Trial (TALENT), enrolled never-smokers who were aged 55 to 75 years [9]. Similarly, other observational studies in never-smokers have imposed an upper age limit of 75 years [10,11,12].

This exclusion is especially concerning given the high burden of disease in older adults: 37% of lung cancer cases occur in those aged 75–84, and 9% in those aged ≥85 [13]. Despite this, older age groups have largely been excluded from previous screening trials, and studies focusing specifically on lung cancer screening in elderly populations remain scarce. One study reported that LDCT screening in individuals aged from 70 to 80 increased the detection of early-stage lung cancer; however, this study focused only on those under 80 years of age [14]. The limited inclusion of very elderly patients in screening trials is often justified by concerns about their lower likelihood of receiving curative treatment, their reduced life expectancy, increased comorbidities, frailty, and the potential diminished benefit of screening in this population [15,16,17,18,19].

Recent guidelines are beginning to reflect a shift toward more individualized screening strategies. The National Comprehensive Cancer Network (NCCN) has removed its upper age limits for lung cancer screening, emphasizing clinical judgment and individualized risk assessment [13]. In contrast, other major guidelines—such as those from the U.S. Preventive Services Task Force (USPSTF) and the Korean national guidelines—continue to impose age-based cutoffs at 80 years and 74 years, respectively [20,21,22].

As the global population ages rapidly, there is an urgent need to reconsider lung cancer screening approaches for older adults. Emerging evidence suggests that many older patients remain eligible for and benefit from curative-intent treatment [23,24,25,26,27,28]. Therefore, excluding them from screening solely based on age may be unwarranted. Furthermore, recent data indicate that nearly two-thirds of newly diagnosed lung cancer cases in the United States do not meet current USPSTF screening criteria, highlighting the need to refine and possibly expand eligibility guidelines [29]. This growing recognition calls for a more nuanced identification of high-risk populations beyond traditional age cutoffs [13].

While LDCT screening has been extensively studied in smoking and non-smoking populations, evidence that specifically focuses on elderly individuals remains scarce. This demographic, however, bears a disproportionate burden of lung cancer and is often excluded from guideline-based screening. Instead of rigid age-based restrictions, screening strategies that account for biological risk and functional status are needed [30]. We hypothesize that a single round of LDCT screening in an elderly, predominantly male, real-world cohort will yield a lung cancer detection rate that is comparable to or exceeds rates reported in RCTs while maintaining a favorable early-stage detection rate, which would suggest a clinical benefit despite patients’ advanced age and comorbidities. This study aims to fill this evidence gap by evaluating lung cancer detection in men aged 70 years and older following this screening intervention. Specifically, we assessed the prevalence and characteristics of pulmonary nodules, the lung cancer detection rate resulting from follow-up, and the stage at diagnosis to explore the potential benefits of screening in this older demographic.

## 2. Materials and Methods

This retrospective single-center cohort study was conducted at the Veterans Health Service Medical Center in Seoul, Republic of Korea. We included male patients aged 70 years or older who underwent their first LDCT scan for lung cancer screening between January 2012 and June 2020.

Inclusion criteria were as follows: (1) age ≥70 years at the time of screening; (2) no symptoms suggestive of lung cancer (e.g., chronic cough, hemoptysis, unexplained weight loss, or dyspnea); (3) underwent LDCT for lung cancer screening either at their own request or upon the recommendation of a physician as part of routine health screening.

In the Korean healthcare system, opportunistic LDCT screening is available for individuals regardless of guideline-defined eligibility (e.g., smoking history). Therefore, even patients who do not meet formal screening criteria—such as never-smokers—may undergo LDCT if they or their physician wish to pursue early detection.

The exclusion criteria were as follows: (1) Prior imaging abnormalities suspicious for cancer that prompted LDCT (e.g., suspicious nodules or consolidations on chest X-ray or CT). (2) Symptoms suspicious for lung cancer present at the time of LDCT. (3) History of prior lung cancer. (4) Missing data on smoking status.

Comorbidity information including chronic obstructive pulmonary disease (COPD) and radiographic emphysema was collected from electronic medical records and radiology reports. Final inclusion flow is illustrated in Figure 1.

Follow-up data were obtained until either loss to follow-up or the study cutoff date (30 September 2021), whichever occurred first. Baseline demographic and clinical characteristics, including age, sex, body mass index (BMI), smoking status (current/former/never), and COPD diagnosis, were assessed using electronic health records.

A preliminary evaluation of the initial low-dose chest CT images was conducted, with a focus on the identification of lung nodules or the presence of lung cancer. Lung nodules were defined as non-calcified lumps with a diameter of at least 4 mm [2,31]. The Lung Imaging Reporting and Data System (Lung-RADS) was used to classify and evaluate these nodules [32]. According to Lung-RADS version 1.1, Category 3 indicates probably benign nodules with a malignancy risk of less than 2%, typically requiring follow-up imaging at 6 months, while Category 4 indicates a higher suspicion of malignancy (>5%), warranting further diagnostic evaluation. Nodules classified as Lung-RADS 3 or 4 were defined as positive nodules [33,34,35]. Patients with a positive lung nodule on initial LDCT were recommended to undergo subsequent LDCT follow-up or intensive lung cancer work-up according to the recommendations of Lung-RADS [33]. In addition, all detected nodules were classified based on their radiographic appearance into solid, part-solid, or ground-glass opacity (GGO) types, and any changes in Lung-RADS categories over time were monitored during follow-up LDCT examinations.

The diagnosis of lung cancer in this study was primarily confirmed through histological diagnosis. However, for patients with a systemic condition classified as Eastern Cooperative Oncology Group (ECOG) performance status 3 or higher, for whom biopsy was deemed infeasible, diagnosis was established based on serial follow-up CT imaging and multidisciplinary evaluation involving pulmonologists and radiologists. The interval between initial screening and lung cancer diagnosis was determined, and cancer characteristics—including diagnostic methods, histopathological classification, stage, and treatment modalities—were analyzed. Diagnostic and therapeutic decisions were made by the attending specialists and, when necessary, confirmed by a multidisciplinary team.

In order to examine the consistency of the findings in a clinically relevant high-risk group, we conducted a sensitivity analysis restricted to smokers. Comparative data from never-smokers were reported for completeness, although formal inference was limited due to the small sample size and event rate.

### 2.1. Statistical Analyses

Categorical variables were presented as numbers and percentages. Differences between groups were assessed using Pearson’s chi-squared test or Fisher’s exact test, as appropriate. Continuous variables with a normal distribution were presented as means ± standard deviations and compared using Student’s *t*-test. Continuous variables that were not normally distributed were reported as median with interquartile range (IQR; 25th–75th percentiles) and compared using the Mann–Whitney U test. Lung cancer detection rate was expressed as a percentage. To test the hypothesis that increasing age is associated with a higher likelihood of detecting biologically aggressive or clinically actionable lung cancers, participants were stratified into five age groups (70–74, 75–79, 80–84, 85–89, and ≥90 years), and subgroup analyses were performed to evaluate trends in the detection rates, nodule types, stage at diagnosis, and treatment patterns. We used the Cochran–Armitage test to analyze trends in lung cancer detection according to age group and visualized the results using line graphs. Differences in the interval from screening to lung cancer diagnosis across age groups were analyzed using the Kruskal–Wallis test. All *p* values were two-sided, and a *p* value of <0.05 was considered statistically significant. All analyses were performed using Stata 17.0 (Stata Corp., College Station, TX, USA).

### 2.2. Ethical Considerations

This study was approved by the Institutional Review Board of the Veteran Healthcare Service Medical Center (Approval No. BOHUN 2023-02-004-003). The requirement for informed consent was waived due to the retrospective nature of this study.

## 3. Results

### 3.1. Characteristics of the Screened Populations

A total of 1409 participants aged 70 years or older underwent LDCT screening. Initially, 7836 individuals were assessed for eligibility; of these, 45 with a history of lung cancer, 6356 who underwent LDCT due to cancer-suspected symptoms or prior imaging findings, and 26 with missing smoking-related data were excluded (Figure 1). The mean age was 74.2 ± 4.9 years, with 959 (68.1%) being aged 70–74 years, 235 (16.7%) being aged 75–79 years, 130 (9.2%) being aged 80–84 years, 76 (5.4%) being aged 85–89 years, and 9 (0.6%) being aged ≥90 years. All participants were male, and the mean BMI was 24.4 ± 3.3 kg/m^2^. COPD was present in 41.7% of the participants. A total of 449 (31.9%) were current smokers, 855 (60.7%) were former smokers, and 105 (7.5%) were never-smokers, with the median smoking history among the smokers being 40 pack-years (Table 1). The median follow-up duration for the entire cohort was 3.6 years (IQR, 2.6–5.8). On average, the participants underwent 2.8 follow-up CT scans. Compared to previous LDCT trials such as NLST [2,36], NELSON [3,37], and UKLS [5], our cohort was notably older, entirely male, and included never-smokers (Appendix A).

### 3.2. Observed Lung Nodule’s Characteristics

Lung nodules were identified in 786 participants (55.8%), with 179 cases (22.8%) being classified as positive (Lung-RADS category ≥ 3). The positive nodules were significantly larger than the non-positive nodules (12.6 ± 10.4 mm vs. 6.6 ± 6.2 mm, *p* < 0.001), more likely to be part-solid (27.9% vs. 8.3%, *p* < 0.001), and predominantly solitary (65.9%, *p* < 0.001). Among the positive findings, the distribution of Lung-RADS categories was as follows: 3 (53.1%), 4A (30.7%), 4B (12.3%), and 4X (3.9%) (Table 2). All nodules classified as Lung-RADS 4X were confirmed as malignant, which underscores the strong diagnostic utility of this classification system. Appendix A further illustrates that although the majority of nodules were classified as Lung-RADS 2, lung cancer detection rates increased markedly with higher categories—2.1% in category 3, 16.4% in 4A, 36.4% in 4B, and 100% in 4X. (Representative CT images of cancerous and non-cancerous nodules detected during screening are presented in Figure 2.

Among the 786 detected nodules, the majority (n = 755, 96.1%) were ultimately non-cancerous. These benign nodules were typically small (mean size: 5.8 ± 3.6 mm), with most being classified as solid (81.5%), followed by the ground-glass opacity (11.0%) and part-solid (7.6%), types. Half of the non-cancerous nodules (50.6%) were solitary, while the other half appeared as multiple lesions. Based on Lung-RADS categorization, 79.7% were assessed as category 2, which indicates minimal concern for malignancy, and only a small proportion fell into higher-risk categories: 12.3% in category 3, 6.1% in 4A, and 1.9% in 4B. None of the non-cancerous nodules were classified as 4X. Regarding the follow-up outcomes, 9.1% of these nodules disappeared and 11.1% decreased in size, which is consistent with a benign natural course. However, 4.9% demonstrated interval growth, which underscores the importance of radiological surveillance, even for nodules that are initially assessed as low risk (Appendix A).

### 3.3. Lung Cancer Detection in Initial Lung Cancer Screening CT

Among the 1409 participants who underwent baseline LDCT screening, 31 (2.2%) were diagnosed with lung cancer (Table 3). The detection increased with age from 1.7% in those aged 75–79 to 11.1% in those aged 90 and above, although this trend was not statistically significant (*p* = 0.146) (Table 3, Figure 3).

By smoking status, the current smokers had the highest detection rate (3.6%), followed by the never-smokers (1.9%) and former smokers (1.5%; *p* = 0.056). The participants with emphysema showed a significantly higher detection rate than those without (3.1% vs. 1.2%, *p* = 0.018), whereas no significant difference was found by COPD status (2.6% vs. 2.0%, *p* = 0.447) (Table 3).

Lung cancer detection was strongly correlated with the Lung-RADS category: 0.8% for category 2, 2.1% for category 3, 16.4% for 4A, 36.4% for 4B, and 100% for 4X (7/7 patients).

The lung cancer detection rates varied significantly by initial nodule type: 3.5% for solid nodules (22/637), 12.3% for part-solid nodules (8/65), and 1.2% for ground-glass opacity nodules (1/84) (*p* = 0.001) (Table 3). Most of the cancers originated from solid nodules (71%), followed by the part-solid (25.8%) and ground-glass opacity nodules (3.2%). The malignant nodules were significantly larger than the non-cancerous nodules (25.0 ± 17.9 mm vs. 5.8 ± 3.6 mm; *p* < 0.001) (Appendix A). The detection rates by age group and clinical subgroups are summarized in Table 4. Among the patients who were diagnosed with lung cancer, the proportion of solid nodules increased with age: from 60.0% in the 70–75 group to 100.0% in those aged ≥80 years. In contrast, subsolid or GGO nodules were observed only in younger age groups.

The detection rate of 2.2% in this study exceeds those reported in the NLST (1.0%), the first round of the NELSON trial (0.9%), the UKLS (2.1%), and the I-ELCAP study [8,36,37,38] (Table 5). In comparison with other major lung cancer screening studies, the proportion of positive lung nodules in our study (12.7%) was similar to that reported in the I-ELCAP (13.3%) and lower than that of the NLST (27.3%), but higher than that reported in the NELSON trial (2.6%). The false positive rate in our cohort (82.7%) was lower than that reported in the NLST (94.3%) but higher than that reported in the NELSON trial (56.5%). Consistent with findings from other screening programs, adenocarcinoma was the most commonly diagnosed lung cancer type. Although stage I lung cancer was the most frequently detected stage in our study (48.4%), this proportion was lower compared to the I-ELCAP study (84.9%) (Table 5).

### 3.4. Time Interval Between Initial Screening CT and Lung Cancer Diagnosis

The median time from baseline screening to lung cancer diagnosis was 9.3 months (IQR 1.5–15.0). The patients diagnosed with lung cancer underwent an average of 3.2 CT scans, compared to 2.8 scans in those without a cancer diagnosis, with there being no significant difference between the groups (*p* = 0.165). The median interval from screening to lung cancer diagnosis was notably shorter in the oldest age groups—1.1 months for individuals aged 75–80 and 1.3 months for those aged ≥90, compared to 10.9 months in the 70–75 age group (Table 6). However, the differences across age groups were not statistically significant (*p* = 0.132).

### 3.5. Lung Cancer Characteristics

Table 6 summarizes the characteristics and treatment patterns of the lung cancer cases detected through screening. Surgical biopsy was the most common diagnostic method (13 patients, 41.9%), followed by bronchoscopy (10 patients, 32.4%) and percutaneous needle biopsy (4 patients, 12.9%). One patient (3.2%) was diagnosed through biopsy at another site, and three patients (9.7%) did not undergo biopsy due to poor medical condition.

Histologically, adenocarcinoma was the most common type of cancer (14 patients, 45.2%), followed by squamous cell carcinoma (12 patients, 38.7%). Small-cell lung cancer and the other NSCLC types were each diagnosed in one patient (3.2%), and three cases (9.7%) had an unknown histology. Nearly half of the patients (n = 15, 48.4%) were diagnosed at stage I; the remainder were diagnosed at stage II (n = 4, 12.9%), stage III (n = 5, 16.1%), and stage IV (n = 7, 22.6%).

Surgical resection was the most frequent initial treatment (n = 17, 54.8%), followed by palliative chemotherapy (n = 7, 22.6%) and definitive radiation therapy (n = 3, 9.7%). Concurrent chemoradiation therapy (CCRT) was given to two patients (6.5%), and one patient (3.2%) received best supportive care. The appropriate treatment was defined as therapy consistent with the NCCN guidelines based on cancer stage. Overall, 28 patients (90.3%) received treatment that was appropriate for their cancer stage, which indicates the adherence to clinical guidelines in this screened cohort.

### 3.6. Sensitivity Analysis in Smokers and Comparative Findings in Never Smokers

In the sensitivity analysis restricted to smokers, the trends in the lung cancer detection rates, cancer stage, and treatment patterns were generally consistent with those observed in the overall cohort. Among the 1304 smokers (mean age 74.2 years), the prevalence of COPD was 43.2% (*p* < 0.001 compared to never smokers). Lung nodules were detected in 724 participants (55.5%), with the mean nodule size being 6.6 mm. A total of 163 participants (12.5%) had positive nodules (≥6 mm), with the mean diameter being 12.5 mm. Lung cancer was diagnosed in 29 smokers (2.2%), with 17 (58.6%) being identified at stage I or II. The most common histologic subtypes were adenocarcinoma (n = 12) and squamous cell carcinoma (n = 12). Fifteen patients (51.7%) underwent surgery, and 26 (89.7%) received treatment that was appropriate for their cancer stage.

For comparison, data from 105 never-smokers were also presented. This group was slightly older (mean age 75.5 years) and had a lower prevalence of COPD (23.8%). Two cases of lung cancer (1.9%) were detected, both of which were stage I adenocarcinomas that were managed with surgical resection. No statistically significant difference in lung cancer detection rates was observed between the smokers and never-smokers (*p* = 0.830) (Table 7).

## 4. Discussion

While major RCTs have established the efficacy of lung cancer screening, our study provides crucial real-world data on a largely underexplored population: elderly individuals (≥ 70 years) who often fall outside strict guideline-defined eligibility. This opportunistic screening context offers a unique perspective, filling a significant knowledge gap by demonstrating the feasibility and actual outcomes of LDCT in a demographic facing an increasing lung cancer burden but which is frequently excluded from large-scale, controlled studies. Our findings thus contribute to a nuanced understanding of the screening effectiveness and challenges specifically within older, diverse cohorts, moving beyond the idealized settings of clinical trials.

In this retrospective cohort study of elderly male smokers aged 70 years or older, a single LDCT screening detected lung nodules in 55.8% of the participants, with 12.7% being identified as positive nodules. With appropriate follow-up, the screening yielded a lung cancer detection rate of 2.2%, and the majority of detected cases (61.3%) were diagnosed at an early stage (I or II). Notably, over 60% of these cancers were treated with curative surgery, which underscores the clinical value of early detection in this underrepresented, high-risk group. These findings suggest that even a one-time LDCT screening can provide meaningful diagnostic and therapeutic benefits for these elderly individuals.

While the benefits of annual LDCT screening are well established, our findings suggest that even a single screening round may provide meaningful benefits in select elderly populations—particularly when repeated screening is limited by life expectancy, comorbidities, or resource constraints. This approach may be especially relevant in settings where full-scale programs are not feasible. Although our study does not introduce new biomarkers or imaging tools, it addresses a critical evidence gap by demonstrating the feasibility and potential impact of screening in older adults. Future research should incorporate objective metrics—such as frailty indices, ECOG performance status, and long-term outcomes—to better define which elderly individuals are most likely to benefit and to guide optimal screening strategies in this high-risk group.

More than half (55.9%) of our cohort had lung nodules ≥4 mm—substantially higher than the 27.3% reported in the NLST using the same threshold [36], which likely reflects the increased prevalence of benign nodules associated with aging [39,40,41,42]. Chen et al. reported a 30% prevalence of nodules ≥4 mm in a general screening cohort, supporting the notion that nodule burden increases with age [39]. Comparison with the ImaLife cohort [41], which evaluated solid nodules volumetrically in a Northern European nonsmoking population, revealed that both studies yielded similar results. Among men aged ≥70, ImaLife reported a prevalence of 56.1–60.7% for nodules ≥30 mm^3^ and 17.3–24.4% for nodules ≥100 mm^3^, assuming spherical nodules, where 30 mm^3^ and 100 mm^3^ correspond to approximate diameters of ~4 mm and ~6 mm, respectively. Although this approximation may be less accurate for irregularly shaped nodules, it still supports the observation that our older, predominantly smoking Asian cohort exhibits a similar burden of significant nodules to Western non-smoking populations, which suggests that aging itself may play a larger role in nodule burden than previously assumed (Appendix A). Given this high prevalence, a larger number of nodules require radiologic surveillance, which would inevitably increase the burden of follow-up evaluation in older adults. This has important implications for healthcare utilization and patient anxiety—factors that were not directly assessed in this study but which should be addressed in future work [43,44,45].

Furthermore, our findings suggest that participants without prior imaging and those presenting with larger nodules at baseline had a higher likelihood of lung cancer diagnosis. This underscores the need for individualized follow-up strategies after the initial LDCT. Tailored surveillance intervals—particularly for individuals with no imaging history or with newly identified large nodules—may enhance early detection, even in single-screen settings.

Although not statistically significant, we observed a trend toward a shorter time from screening to lung cancer diagnosis with increasing age. This may be due to the predominance of solid nodules in older patients, which are more likely to be invasive and prompt faster diagnostic work-up than subsolid or ground-glass lesions, which often undergo surveillance. In our cohort, all cancers in patients aged ≥80 were linked to solid nodules, whereas younger patients had cancers arising from both solid and subsolid nodules. Clinical decisions to prioritize suspicious lesions in very elderly patients and to avoid the aggressive evaluation of indolent lesions like ground-glass opacities—considering limited life expectancy—may contribute to the shorter diagnostic intervals seen in the oldest group.

Our lung cancer detection rate of 2.2% was higher than those of the first screening rounds of the NLST (1.0%) and NELSON (0.9%), and comparable to that of UKLS (2.1%) (Table 4) [8,36,37]. This notably higher detection rate is a critical finding, primarily reflecting the advanced age and consequently elevated baseline risk within our cohort. Given that age is a well-established and powerful independent risk factor for lung cancer incidence, the increased detection rate observed in our study strongly underscores the substantial burden of undiagnosed lung cancer in this older population [46].

The issue of overdiagnosis remains a significant concern in lung cancer screening. However, our data suggest that this issue may be less pronounced in our cohort. First, the histopathologic spectrum showed a predominance of solid and invasive histology (84%), with minimal detection of indolent ground-glass nodules (only one GGO cancer). Notably, no cases of adenocarcinoma in situ (AIS)—a subtype frequently linked to overdiagnosis [47] due to its indolent nature—were observed in our cohort. This absence suggests that our screening predominantly identified clinically significant cancers. Second, the time from detection to diagnosis was relatively short, especially in the older subgroups, which supports the likelihood of aggressive disease biology. Collectively, these findings suggest that LDCT screening in older male adults primarily detects biologically actionable disease, which minimizes the risk of overdiagnosis. Nonetheless, we acknowledge that our retrospective design limits the ability to definitively quantify overdiagnosis.

In terms of the stage distribution, 61% of the detected lung cancers in our study were diagnosed at early stages (stage I–II), a proportion that is comparable or higher to those of DANTE (67.8%) [47], ITALUNG (43%) [4], and Depiscan (38%) [48], but somewhat lower than those of UKLS (83%) [8], the NLST (70%) [2], and NELSON (68%) [3]. This intermediate early-stage detection rate suggests that LDCT screening remains effective in identifying potentially curable cancers, even in an older, high-risk population. Conversely, our cohort had a higher proportion of stage IV cancers (23%) compared to the NLST (13%) [2], NELSON (9%) [3], UKLS (5%) [8], DANTE (14%) [48], and DEPISCAN (13%) [49], but a lower proportion than that reported in ITALUNG (36%) [4]. This likely reflects the advanced age of our study population and the absence of prior screening, which may have contributed to delayed diagnosis. Nevertheless, when compared to national lung cancer data in Korea—where 45.0% of cases are diagnosed at stage IV and only 36.9% at early stages [50]—our results show a notably more favorable stage distribution. Recent study from Feng et al. highlighted that reductions in late-stage diagnoses through screening are strongly linked to improved survival [51], reinforcing the potential benefit of LDCT even in older populations. This suggests that LDCT screening in elderly male smokers can facilitate earlier detection and improve treatment opportunities.

In terms of histologic subtype, adenocarcinoma was the most commonly diagnosed type in our cohort (41%), similar to in other LDCT trials where adenocarcinoma predominated—such as the NLST (55%, including bronchioloalveolar carcinoma [BAC]) [2], NELSON (61%) [3], UKLS (60%) [8], DANTE (61%, including BAC) [48], Depiscan (63%) [49], and ITALUNG (43%) [4]. This pattern aligns with the general trend observed in screened populations, as adenocarcinomas tend to be more detectable. A notable distinction in our study is the relatively high proportion of squamous cell carcinoma, which accounted for 41% of the diagnosed cases—substantially higher than in the NLST (21%) [2], NELSON (19%) [3], UKLS (29%) [8], DANTE (28%) [48], and ITALUNG (21%) [4]. This may reflect our study population’s older age, higher smoking burden, and male predominance, as squamous cell carcinoma is more strongly associated with smoking and male sex [52,53,54].

Our study suggests that LDCT screening in older adults can lead to early detection and timely, stage-appropriate treatment. Nearly half of the patients underwent curative surgery for early-stage disease, which supports the potential of screening to improve outcomes in the elderly. These findings highlight the importance of considering patients’ physiologic status and individual risk, rather than age alone, when making screening decisions.

In line with previous LDCT screening studies, we also analyzed the outcomes among smokers. The lung cancer detection rate in this group was 2.2%, with a higher proportion of advanced-stage diagnoses compared to never-smokers. In contrast, all cancers in never-smokers were early-stage adenocarcinomas that were treated with curative surgery, which is consistent with prior findings that never-smokers tend to develop indolent tumors [9,10,55]. Notably, the proportion of patients receiving stage-appropriate treatment was high in both the smokers and never-smokers (89.7% in smokers vs. 100% in never-smokers). In addition, the presence of emphysema was significantly associated with increased lung cancer detection, which is consistent with findings from the NLST, MILD, and other studies [2,3,5,56]. In contrast, COPD showed a non-significant trend toward higher detection, which suggested that emphysema may be a more sensitive marker of underlying risk in our LDCT screening populations.

Our findings highlight several distinct features of lung cancer screening in the elderly population. First, lung cancers were predominantly detected in solid nodules, particularly among individuals aged 80 years or older. This contrasts with previous trials, where subsolid nodules—including part-solid and ground-glass nodules—contributed significantly to the cancer diagnoses [57,58,59]. Second, squamous cell carcinoma was relatively more common in this elderly cohort, accounting for nearly 40% of cases. Third, the majority of lung cancers were diagnosed within three years of the initial screening, which suggests a shorter natural course and supports the utility of a single LDCT with limited follow-up in this age group. These findings imply that lung cancers detected through screening in the elderly may exhibit more aggressive and rapidly progressing behavior, underscoring the need for age-tailored screening strategies.

A key strength of this study is its use of real-world data that are focused on an underrepresented, high-risk group—elderly male smokers—as well as including non-smoker data that are relevant to the Asian population, where non-smoking lung cancer is more prevalent. By analyzing the screening outcomes by age and detailing nodule characteristics and clinical results, we provide practical insights into the effectiveness of LDCT screening in routine clinical settings. Despite concerns about frailty and limited life expectancy, our findings demonstrate that even a single LDCT screen can detect a significant number of treatable lung cancers in this population.

Several limitations should be considered. First, the retrospective, single-center design may limit the generalizability of our findings to broader populations. Second, the age distribution was skewed toward individuals in their 70s, with relatively few participants being aged 90 or older, which limits insights into outcomes among the very elderly. Third, the median follow-up duration after initial LDCT screening was relatively short, potentially limiting the detection of indolent cancers and the evaluation of long-term outcomes. Lastly, due to the retrospective design and limited clinical follow-up data, we could not assess downstream outcomes such as survival, biopsy-related complications, screening-related overdiagnosis, patient anxiety, or treatment decisions. Future prospective studies with control groups and longer follow-up are needed to clarify the long-term benefits and potential harms of LDCT screening in older adults.

Our study contributes unique real-world evidence on the utility of LDCT screening in a demographically older and clinically distinct population, in contrast to previous trials such as the NLST and NELSON, which excluded individuals aged ≥75 years or never-smokers. By including both elderly smokers and never-smokers, our study provides a more comprehensive view of the screening performance across varying risk profiles. Meaningful detection rates and substantial treatment uptake—even among those aged ≥80 years—highlight the feasibility and clinical value of individualized screening strategies in routine practice.

However, caution is still warranted when applying our data to clinical practice, and several limitations should be carefully considered. This cohort, due to the inclusion of some never-smokers, cannot be considered fully representative of the entire smoking population, nor does it represent the full spectrum of never-smokers. Never-smokers were incorporated into our study design to generate much-needed screening data for this underrepresented population in current trials, but their number and the absence of survival rates necessitate careful interpretation of this study’s findings. Furthermore, our study does not provide absolute evidence that lung cancer screening will yield benefits in this specific population. Importantly, the all-male, older study population limits the generalizability of these findings to women and younger individuals. Additionally, the participants were able to visit a clinic for screening, which indicates that they likely represent a healthier and more health-conscious subset of the elderly—particularly the smokers who willing and able to attend a pulmonology clinic—and thus introduces potential selection bias. These factors should be meticulously considered when interpreting detection rates and applying our findings to broader lung cancer screening policies. Therefore, future prospective studies with comprehensive data on both smokers and never-smokers, including long-term survival outcomes, are essential to validate and expand upon our findings.

## 5. Conclusions

This study highlights the significant clinical advantage of a single low-dose CT screening in elderly males, primarily smokers, by effectively identifying a substantial number of early-stage lung cancers, many of which are potentially curable. Early detection in this high-risk population has important clinical implications, supporting the expansion of lung cancer screening guidelines to more inclusively consider elderly smokers. These findings support the implementation of individualized LDCT screening protocols that prioritize patients’ physiologic reserve and risk profile over age, which would ensure that high-risk elderly populations are not unjustly excluded from potentially life-saving early detection.

Future prospective studies with well-designed, risk-adaptive protocols—such as randomized controlled trials or longitudinal cohort studies that incorporate frailty and comorbidity assessments—are essential to evaluate long-term clinical outcomes and optimize screening intervals for this vulnerable group.

## Figures and Tables

**Figure 1 cancers-17-02318-f001:**
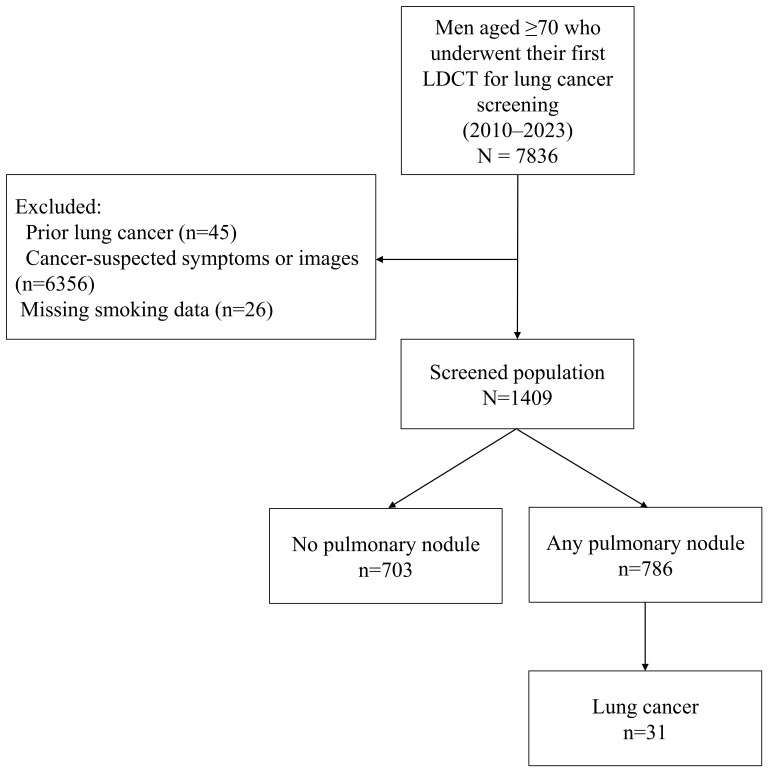
Flow chart of participants.

**Figure 2 cancers-17-02318-f002:**
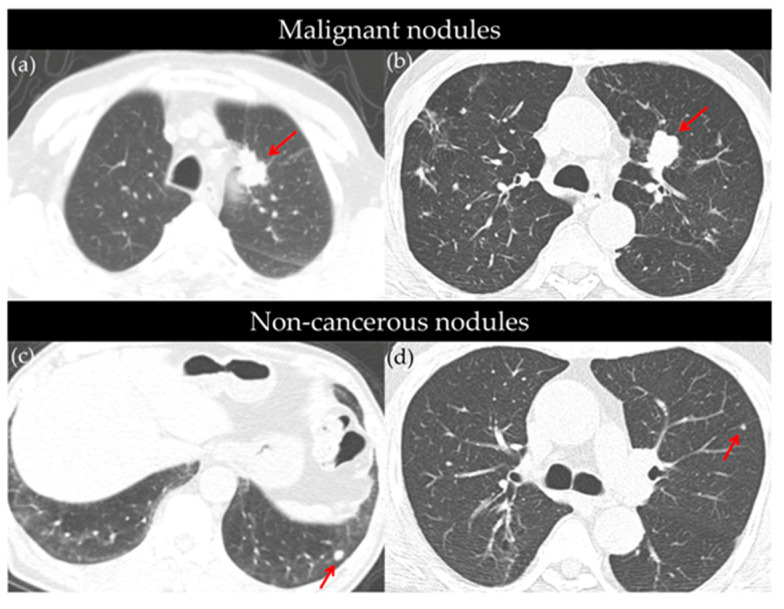
Representative axial CT images of cancerous (**a,b**) and non-cancerous (**c,d**) pulmonary nodules identified through low-dose CT screening. (**a**) Approximately 3 cm mass in the paramediastinal area of the left upper lobe (LUL) anterior segment (red arrow), diagnosed as adenocarcinoma. (**b**) A 2.4 cm sized nodular opacity in the LUL (red arrow), pathologically confirmed as squamous cell carcinoma. (**c**) An 8 mm sized nodule in the left lower lobe (LLL) (red arrow), with no significant size change over three serial follow-up CTs; no cancer diagnosis over a 10-year observation period. (**d**) A 6 mm nodule in the LUL (red arrow) followed-up with two additional CTs; remained stable, with no lung cancer diagnosis over an 8-year follow-up period.

**Figure 3 cancers-17-02318-f003:**
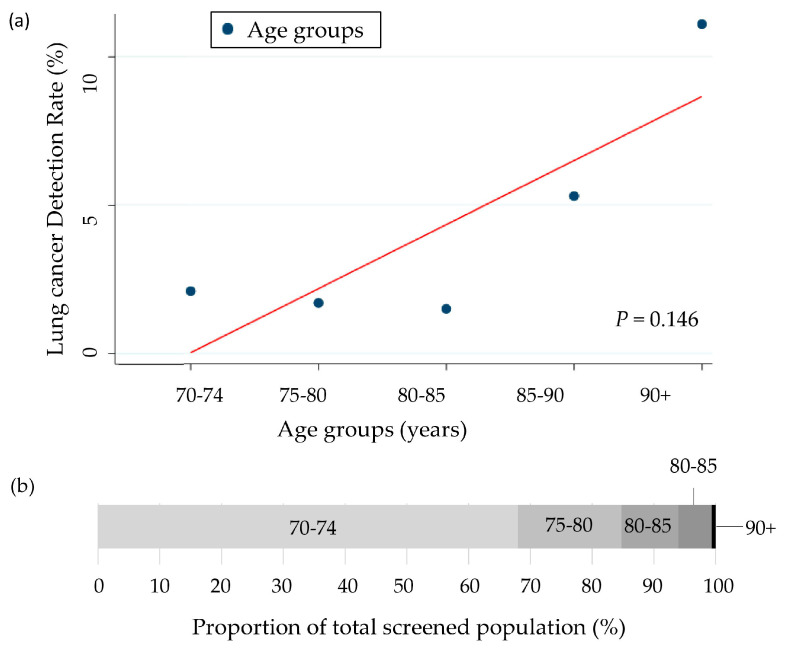
(**a**) Lung cancer detection rates across age groups with trend line (Cochran–Armitage test). (**b**) Proportion of total screened population by age group (%).

**Table 1 cancers-17-02318-t001:** Baseline characteristics of participants who received LDCT screening.

Characteristics	Total Participants (n = 1409)
Male sex, no. (%)	1409 (100)
Age, years, mean	74.2 ± 4.9
Age, years, range	
70–74	959 (68.1)
75–79	235 (16.7)
80–84	130 (9.2)
85–89	76 (5.4)
90 or older	9 (0.6)
COPD, no. (%)	588 (41.7)
BMI, kg/m^2^, mean	24.4 ± 3.3
Smoking status, no. (%)	
Current smoker	449 (31.9)
Formal smoker	855 (60.7)
Never smoker	105 (7.5)
Median pack-years of smoking among smokers	40 [25–50]
Median follow-up duration (years)	3.6 [2.6–5.8]

**Table 2 cancers-17-02318-t002:** Baseline characteristics of lung nodules.

Characteristics	Patients with Any Nodule (n = 786)	Patients with Positive Nodule (n = 179)	*p*-Value
Nodule size (mm)	6.6 ± 6.2	12.6 ± 10.4	<0.001
Nodule type, no. (%)			<0.001
Solid	637 (81.0)	127 (71.0)	
Part-solid	65 (8.3)	50 (27.9)	
Ground-glass opacity	84 (10.7)	2 (1.1)	
Nodules (%)			<0.001
Single	408 (51.9)	118 (65.9)	
Multiple	378 (48.1)	61 (34.1)	
Lung-RADS			<0.001
2	607 (77.2)	0 (0.0)	
3	95 (12.1)	95 (53.1)	
4A	55 (7.0)	55 (30.7)	
4B	22 (2.8)	22 (12.3)	
4X	7 (0.9)	7 (3.9)	

Lung-RADS, Lung Imaging Reporting and Data System.

**Table 3 cancers-17-02318-t003:** Detection of lung cancer by subgroup in a single round of lung cancer screening.

Variables	Screening Population	Patients with Cancer	Cancer Detection (%)
Total screened population	1409	31	2.2
Age group (years)			
70–75	959	20	2.1
75–80	236	4	1.7
80–85	130	2	1.5
85–90	76	4	5.3
90+	9	1	11.1
Smoking history			
Current smoker	449	16	3.6 *
Former smoker	855	13	1.5 *
Never smoker	105	2	1.9 *
COPD	588	15	2.6 †
Non-COPD	821	16	2.0 †
Emphysema	751	23	3.1 ‡
Non-emphysema	658	8	1.2 ‡
Lung-RADS score			
2	607	5	0.8
3	95	2	2.1
4A	55	9	16.4
4B	22	8	36.4
4X	7	7	100
Initial Lung Nodule type			
Solid	637	22	3.5 §
Partial solid	65	8	12.3 §
Ground-glass opacity	84	1	1.2 §

COPD, chronic obstructive pulmonary disease; * *p*-value for the three groups: 0.056, † *p*-value for the two groups: 0.447, ‡ *p*-value for the two groups: 0.018, § *p*-value for the three groups: 0.001.

**Table 4 cancers-17-02318-t004:** Detection of lung cancer by age group and clinical subgroups.

Variables	70–75 n = 959	75–80 n = 236	80–85 n = 130	85–90 n = 76	90– n = 9
Lung cancer in total screened population	20 of 959 (2.1)	4 of 236 (1.7)	2 of 130 (1.5)	4 of 76 (5.3)	1 of 9 (11.1)
Smoking history					
Current smoker	10 of 353 (2.8)	2 of 63 (3.2)	2 of 16 (12.5)	16 of 1 (6.3)	1 of 1 (100)
Former smoker	9 of 547 (1.7)	1 of 151 (0.7)	0 of 96 (0.0)	3 of 53 (5.7)	0 of 8 (0.0)
Never smoker	1 of 59 (1.7)	1 of 21 (4.8)	0 of 18 (0.0)	0 of 7 (0.0)	0 of 0 (0.0)
COPD	11 of 389 (2.8)	1 of 107 (0.9)	2 of 51 (3.9)	1 of 76 (2.8)	0 of 5 (0.0)
Non-COPD	9 of 570 (1.6)	3 of 128 (2.3)	0 of 79 (0.0)	3 of 40 (7.5)	1 of 4 (25.0)
Emphysema	15 of 498 (3.0)	1 of 134 (0.8)	2 of 65 (3.1)	4 of 49 (8.2)	1 of 9 (11.1)
Non-emphysema	5 of 461 (1.1)	3 of 101 (3.0)	0 of 65 (0.0)	0 of 27 (0.0)	0 of 4 (0.0)
Lung-RADS score					
2	4 of 429 (0.9)	0 of 94 (0.0)	0 of 56 (0.0)	1 of 25 (4.0)	0 of 0 (0.0)
3	1 of 61 (1.6)	1 of 23 (4.4)	0 of 6 (0.0)	0 of 5 (0.0)	0 of 0 (0.0)
4A	8 of 33 (24.2)	0 of 8 (0.0)	1 of 12 (8.3)	0 of 2 (0.0)	0 of 1 (0.0)
4B	4 of 12 (33.3)	2 of 2(100.0)	1 of 5 (20.0)	1 of 2 (50.0)	1 of 1 (100.0)
4X	3 of 3 (100.0)	1 of 1 (100.0)	0 of 0(0.0)	2 of 2 (100.0)	-
Initial Lung Nodule type, no. (%)					
Solid	12 of 431 (2.8)	3 of 107 (2.8)	2 of 67 (3.0)	4 of 28 (14.3)	4 of 1 (25.0)
Partial solid	7 of 44 (15.9)	1 of 12 (8.3)	0 of 4 (0.0)	0 of 5 (0.0)	-
Ground-glass opacity	1 of 63 (3.7)	0 of 9 (0.0)	0 of 8 (0.0)	0 of 3 (0.0)	0 of 1 (0.0)

**Table 5 cancers-17-02318-t005:** Positive lung nodules, lung cancer detection, and detected lung cancer characteristics in lung cancer screening programs.

Variables	Current Study n = 1409	NLST Screening Round 1 n = 26,715	NELSON Screening Round 1 n = 7135	UKLS [8] n = 1994	I-ELCAP [38] † (Baseline Screening) n = 31,567
Positive lung nodules	179 (12.7)	7,191 (27.3)	184 (2.6)	522 (26.7) *	4186 (13.3)
Lung cancer detection (n, %)	31 (2.2)	270 (1.0)	66 (0.9)	42 (2.1)	410 (1.3)
False positive	82.7	94.3	56.5	24.6	90.2
Positive predictive value	17.3	3.8	33.7	8.0	9.8
Sensitivity	Not available	93.8	92.5	93.3	Not available
Specificity	Not available	73.4	98.3	75.4	Not available
Negative predictive value	Not available	99.9	99.9	99.8	Not available
Prevalence lung cancer	n = 31	n = 270	n = 66	n = 42	
Histologic type					
Adenocarcinoma	14 (45.2)	156 (57.7)	34 (51.5)	25 (59.5)	263 (75.6) ‡
Squamous cell carcinoma	12 (38.7)	47 (17.4)	11 (16.7)	12 (28.6)	45 (12.9) ‡
Small-cell lung cancer	1 (3.2)	15 (5.6)	1 (1.5)	3 (7.1)	9 (2.6) ‡
Other non-small-cell lung cancer	1 (3.2)	34 (12.6)	13 (19.7)	0 (0.0)	5 (1.4) ‡
Unknown or other	3 (9.7)	18 (6.7)	3 (4.5)	1 (2.4)	26 (7.5) ‡
Stage, no. (%)					
I	15 (48.4)	155 (64.5)	44 (66.7)	27 (64.3)	348 (84.9)
II	4 (12.9)	18 (6.6)	5 (7.6)	8 (19.0)	Not provided
III	5 (16.1)	52 (19.3)	13 (19.7)	5 (11.9)	Not provided
IV	7 (22.6)	41 (15.2)	4 (6.0)	2 (4.8)	Not provided

* Category ≥3 nodules in initial CT, † compared with baseline screening, ‡ histology was available for stage I patients only (n = 348).

**Table 6 cancers-17-02318-t006:** Characteristics and treatment patterns of lung cancer.

Variables	Patients with Lung Cancer (n = 31)
Median interval between baseline screening and diagnosis, months	9.3 [1.5–15.0]
70–75	10.9 [3.7–24.5]
75–80	1.1 [0.6–2.7]
80–85	15.0 [0.5–29.6]
85–90	8.7 [1.8–14.8]
90–	1.3 [1.3–1.3]
Diagnostic methods, no. (%)	
Percutaneous needle biopsy	4 (12.9)
Bronchoscopy	10 (32.4)
Surgery	13 (41.9)
Biopsies performed at other sites	1 (3.2)
Patients who did not undergo biopsy due to poor medical condition, no. (%)	3(9.7)
Histologic type	
Adenocarcinoma	14 (45.2)
Squamous cell carcinoma	12 (38.7)
Small-cell lung cancer	1 (3.2)
Other non-small-cell lung cancer	1 (3.2)
Unknown	3 (9.7)
Stage, no. (%)	
I	15 (48.4)
II	4 (12.9)
III	5 (16.1)
IV	7 (22.6)
Initial treatment, no. (%)	
Surgery	17 (54.8)
CCRT	2 (6.5)
SABR	0 (0.0)
Definite Radiation therapy	3 (9.7)
Palliative chemotherapy	7 (22.6)
Best supportive care	1 (3.2)
Unknown	1 (3.2)
Appropriate treatment according to cancer staging, no. (%)	28 (90.3)

Values were presented as a number (%) and median (interquartile range). SABR, stereotactic ablative radiation therapy; CCRT, concurrent chemoradiation therapy; Data are presented as No. (%), or median (interquartile range).

**Table 7 cancers-17-02318-t007:** Comparison of clinical characteristics, nodule findings, and lung cancer outcomes between smokers and never-smokers.

Characteristic	Smokers (n = 1304)	Never Smokers (n = 105)	*p*-Value
**Demographics**			
Age, years, mean ± SD	74.2 ± 4.9	75.5 ± 4.9	0.006
BMI, kg/m^2^, mean ± SD	24.4 ± 3.3	24.8 ± 3.1	0.246
COPD, no. (%)	563 (43.2)	25 (23.8)	<0.001
Median pack-years	40 [25–50]	–	
Median follow-up duration, years	3.6 [2.8–4.7]	3.6 [2.6–5.7]	0.012
Number of LDCT	2.8 ± 1.9	2.4 ± 1.8	0.024
**Lung nodule findings**			
Any nodule	724 (55.5)	62 (59.1)	0.484
Diameter, mm	6.6 ± 6.3	6.1 ± 4.8	0.543
Nodule type (%)	n = 724	n = 62	0.289
Solid	584 (80.7)	53 (85.5)	
Part-solid	59 (8.2)	6 (9.7)	
Pure GGN	81 (11.2)	3 (4.9)	
Positive nodule ≥ 6 mm	163 (12.5)	16 (15.2)	0.650
Diameter, mm	12.5 ± 10.6	11.3 ± 7.4	0.651
Nodule type (%)	n = 163	n = 16	0.289
Solid	117 (71.8)	10 (62.5)	
Part-solid	44 (27.0)	6 (37.5)	
Pure GGN	2 (1.2)	0 (0.0)	
Nodule multiplicity (%)			0.802
Solitary nodule	107 (65.6)	11 (68.8)	
Multiple nodules	56 (34.4)	5 (31.3)	
Lung-RADS score			0.760
3	85 (52.1)	10 (62.5)	
4A	51 (31.3)	4 (25.0)	
4B	20 (12.3)	2 (12.5)	
4X	7 (4.3)	0 (0.0)	
**Lung cancer diagnosis**			
Lung cancer diagnosed, no. (%)	29 (2.2)	2 (1.9)	0.830
Time to cancer diagnosis, median (mo)	6.7 [1.3–14.9]	6.6 [3.9–5.5]	0.936
Stage at diagnosis (%)			0.516
Stage I	13 (44.8)	2 (100.0)	
Stage II	4 (13.8)	0 (0.0)	
Stage III	5 (17.2)	0 (0.0)	
Stage IV	7 (24.1)	0 (0.0)	
Histologic type (%)			0.628
Adenocarcinoma	12 (41.4)	2 (100%)	
Squamous cell carcinoma	12 (41.4)	0	
Other non-small cell lung cancer	1 (3.5)	0	
Small cell lung cancer	1 (3.5)	0	
Unknown	3 (10.3)	0	
Treatment modality (%)			0.881
Surgery only	15 (51.7)	2 (100)	
CCRT	2 (6.9)	0	
Definite Radiation therapy	3 (10.3)	0	
Palliative chemotherapy	7 (24.1)	0	
Best supportive care	1 (3.4)	0	
Unknown	1 (3.4)	0	
Appropriate treatment according to cancer staging, no. (%)	26 (89.7)	2 (100.0)	0.892

## Data Availability

Data supporting the findings of this study are available from the corresponding author upon request.

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
