# Peer review of "Single-Round LDCT Screening in Men Aged ≥ 70 Years: Prevalence of Pulmonary Nodules and Lung Cancer Detection"

_cancers, 2025, doi:10.3390/cancers17142318_

Round 1
Reviewer 1 Report
Comments and Suggestions for Authors
Single‑Round LDCT Screening in Men Aged ≥ 70 Years: Prevalence of Pulmonary Nodules and Lung Cancer Detection
Abstract:
- Background is clearly stated; consider citing specific data gaps or controversies regarding LDCT utility in elderly populations.
- Methodology is appropriate; including inclusion/exclusion criteria or comorbidity data could strengthen context and applicability.
- Results are concise and informative; consider reporting confidence intervals for prevalence and detection rates to enhance statistical clarity.
- Conclusion effectively highlights potential benefit; a brief mention of harms (e.g., overdiagnosis) would provide a balanced interpretation.
- The high proportion of early-stage diagnoses is compelling; discussing implications for treatment feasibility in elderly could add value.
Introduction:
- Background provides a strong rationale; consider improving grammar and flow for readability and professionalism.
- References are relevant and up-to-date; ensure consistent citation formatting (e.g., spacing, capitalization).
- Clear justification for focusing on age ≥70; consider clarifying why previous studies excluded those over 80.
- Paragraphing and transitions need refinement to improve logical flow between guideline discussions and evidence gaps.
- The final study aim is well-stated; condensing earlier content could sharpen focus and reduce redundancy.
- Please try to address previous studies targeting at non-smoking population, smoking population. However limited studies on old age population.
References:
Understanding East-West differences in subsolid nodules: prevalence and overdiagnosis implications in lung cancer screening YC Chang, YC Hung, YJ Wu, EK Tang, FZ Wu Annals of Medicine 57 (1), 2478321
Toward more effective lung cancer risk stratification to empower screening programs for the Asian nonsmoking population
FZ Wu, YC Chang
Journal of the American College of Radiology 20 (2), 156-161
Materials and Methods:
- Study design and eligibility criteria are clear; consider reorganizing for smoother logical flow between inclusion and exclusion.
- Use of Lung-RADS is appropriate; briefly clarifying classification thresholds for 3 vs. 4 would aid reader understanding.
- Diagnostic criteria for lung cancer are well-defined; clarify how cases without histology were validated.
- Statistical methods are appropriate; ensure consistent formatting of tests and clearly distinguish between parametric and non-parametric analyses.
- Ethical approval is stated appropriately; consider placing IRB details in a separate "Ethical Considerations" subsection for clarity.
Results:
- Participant characteristics are clearly presented; consider summarizing key findings in text for easier comparison with other cohorts.
- Lung nodule stratification by Lung-RADS is thorough; adding visual aids like charts could enhance clarity.
- Cancer detection rate exceeds prior trials; highlight potential reasons (e.g., age, selection criteria) in Discussion.
- Lung-RADS 4X perfectly correlating with malignancy is striking; consider emphasizing its diagnostic utility.
- Subgroup analyses by age, smoking, and emphysema are well done; note limitations of non-significant trends explicitly.
- Time-to-diagnosis analysis is informative; consider discussing clinical implications of short intervals in older age groups.
- Median diagnostic delays across age groups could be visualized better with a table or a Kaplan-Meier curve.
- Histologic breakdown is well detailed; consider highlighting high adenocarcinoma rate in Discussion for context.
- Treatment modalities reflect real-world practice; strong point that 90% received stage-appropriate therapy.
- Clarify how “appropriate treatment” was defined or determined—based on guidelines or expert consensus?
Discussion:
- Strong justification of higher detection rate; consider reinforcing this with adjusted comparisons across trials accounting for age.
- Valuable comparison to ImaLife; specify more clearly any potential limitations in size-to-volume conversion methodology.
- Consider briefly acknowledging the all-male cohort as a limitation affecting generalizability.
- Discussion could be strengthened by suggesting how findings may influence future screening guidelines for those over 70.
- Well-argued support for LDCT in elderly; adding survival outcomes (if available) could further reinforce the clinical benefit.
- Valuable observation on nodule size and prior imaging absence; suggest mentioning implications for surveillance protocols post-baseline.
- Solid rationale for age-inclusive screening; clarify how frailty was assessed or how future studies might address it.
- Limitations are clearly acknowledged; suggest separating the final paragraph for improved clarity and emphasis on generalizability concerns.
- Please try to address the difference between old aged or general population in lung cancer characters , growth pattern, natural course.
References: Natural history of persistent pulmonary subsolid nodules: long-term observation of different interval growth
EK Tang, CS Chen, CC Wu, MT Wu, TL Yang, HL Liang, HT Hsu, FZ Wu
Heart, Lung and Circulation 28 (11), 1747-1754
Conclusions:
- The conclusion is well-stated; consider briefly reiterating the potential impact on screening guidelines for elderly smokers.
- Statement on individualized screening is valuable; future research directions could benefit from specifying proposed study designs.
- Author contributions are clearly outlined; minor formatting inconsistencies (e.g., punctuation, spacing) should be corrected for clarity.
- The supplementary materials reference is useful; ensure the link format complies with the journal’s final submission guidelines.
- Consider rephrasing the opening sentence slightly to better emphasize the clinical significance of early-stage detection.
Comments on the Quality of English Language
Single‑Round LDCT Screening in Men Aged ≥ 70 Years: Prevalence of Pulmonary Nodules and Lung Cancer Detection
Abstract:
- Background is clearly stated; consider citing specific data gaps or controversies regarding LDCT utility in elderly populations.
- Methodology is appropriate; including inclusion/exclusion criteria or comorbidity data could strengthen context and applicability.
- Results are concise and informative; consider reporting confidence intervals for prevalence and detection rates to enhance statistical clarity.
- Conclusion effectively highlights potential benefit; a brief mention of harms (e.g., overdiagnosis) would provide a balanced interpretation.
- The high proportion of early-stage diagnoses is compelling; discussing implications for treatment feasibility in elderly could add value.
Introduction:
- Background provides a strong rationale; consider improving grammar and flow for readability and professionalism.
- References are relevant and up-to-date; ensure consistent citation formatting (e.g., spacing, capitalization).
- Clear justification for focusing on age ≥70; consider clarifying why previous studies excluded those over 80.
- Paragraphing and transitions need refinement to improve logical flow between guideline discussions and evidence gaps.
- The final study aim is well-stated; condensing earlier content could sharpen focus and reduce redundancy.
- Please try to address previous studies targeting at non-smoking population, smoking population. However limited studies on old age population.
References:
Understanding East-West differences in subsolid nodules: prevalence and overdiagnosis implications in lung cancer screening YC Chang, YC Hung, YJ Wu, EK Tang, FZ Wu Annals of Medicine 57 (1), 2478321
Toward more effective lung cancer risk stratification to empower screening programs for the Asian nonsmoking population
FZ Wu, YC Chang
Journal of the American College of Radiology 20 (2), 156-161
Materials and Methods:
- Study design and eligibility criteria are clear; consider reorganizing for smoother logical flow between inclusion and exclusion.
- Use of Lung-RADS is appropriate; briefly clarifying classification thresholds for 3 vs. 4 would aid reader understanding.
- Diagnostic criteria for lung cancer are well-defined; clarify how cases without histology were validated.
- Statistical methods are appropriate; ensure consistent formatting of tests and clearly distinguish between parametric and non-parametric analyses.
- Ethical approval is stated appropriately; consider placing IRB details in a separate "Ethical Considerations" subsection for clarity.
Results:
- Participant characteristics are clearly presented; consider summarizing key findings in text for easier comparison with other cohorts.
- Lung nodule stratification by Lung-RADS is thorough; adding visual aids like charts could enhance clarity.
- Cancer detection rate exceeds prior trials; highlight potential reasons (e.g., age, selection criteria) in Discussion.
- Lung-RADS 4X perfectly correlating with malignancy is striking; consider emphasizing its diagnostic utility.
- Subgroup analyses by age, smoking, and emphysema are well done; note limitations of non-significant trends explicitly.
- Time-to-diagnosis analysis is informative; consider discussing clinical implications of short intervals in older age groups.
- Median diagnostic delays across age groups could be visualized better with a table or a Kaplan-Meier curve.
- Histologic breakdown is well detailed; consider highlighting high adenocarcinoma rate in Discussion for context.
- Treatment modalities reflect real-world practice; strong point that 90% received stage-appropriate therapy.
- Clarify how “appropriate treatment” was defined or determined—based on guidelines or expert consensus?
Discussion:
- Strong justification of higher detection rate; consider reinforcing this with adjusted comparisons across trials accounting for age.
- Valuable comparison to ImaLife; specify more clearly any potential limitations in size-to-volume conversion methodology.
- Consider briefly acknowledging the all-male cohort as a limitation affecting generalizability.
- Discussion could be strengthened by suggesting how findings may influence future screening guidelines for those over 70.
- Well-argued support for LDCT in elderly; adding survival outcomes (if available) could further reinforce the clinical benefit.
- Valuable observation on nodule size and prior imaging absence; suggest mentioning implications for surveillance protocols post-baseline.
- Solid rationale for age-inclusive screening; clarify how frailty was assessed or how future studies might address it.
- Limitations are clearly acknowledged; suggest separating the final paragraph for improved clarity and emphasis on generalizability concerns.
- Please try to address the difference between old aged or general population in lung cancer characters , growth pattern, natural course.
References: Natural history of persistent pulmonary subsolid nodules: long-term observation of different interval growth
EK Tang, CS Chen, CC Wu, MT Wu, TL Yang, HL Liang, HT Hsu, FZ Wu
Heart, Lung and Circulation 28 (11), 1747-1754
Conclusions:
- The conclusion is well-stated; consider briefly reiterating the potential impact on screening guidelines for elderly smokers.
- Statement on individualized screening is valuable; future research directions could benefit from specifying proposed study designs.
- Author contributions are clearly outlined; minor formatting inconsistencies (e.g., punctuation, spacing) should be corrected for clarity.
- The supplementary materials reference is useful; ensure the link format complies with the journal’s final submission guidelines.
- Consider rephrasing the opening sentence slightly to better emphasize the clinical significance of early-stage detection.
Reviewer 2 Report
Comments and Suggestions for Authors
This manuscript presents a retrospective analysis of a single-round LDCT screening program in men aged ≥70 years. While the subject is relevant, the scientific contribution is limited due to several significant concerns:
-
The main findings are largely confirmatory, reiterating nodule prevalence and cancer detection rates already reported in prior major studies such as NLST, DANTE, MILD, and COSMOS, without advancing new clinical insights or recommendations.
-
The subgroup analyses (e.g., age, pack-years, years since cessation) are standard and do not yield novel or clinically actionable stratifications.
-
A major issue is the inclusion of 7.5% non-smokers in the screening cohort. This contradicts the fundamental rationale of LDCT screening, which is based on risk stratification in heavy smokers or former smokers, as established in all major guidelines. The lack of a clear justification or discussion around this point undermines the study’s internal validity and generalizability.
-
The inclusion and exclusion criteria are poorly defined. It is unclear how asymptomatic status was verified, and whether prior imaging, cancer history, or comorbidities were exclusionary.
-
The manuscript lacks references to key European trials such as BioMILD, UKLS, NELSON (full data), and DEPISCAN, which limits the depth of the discussion.
-
The potential harms of LDCT screening in older populations—particularly false positives, overdiagnosis, follow-up burden, and treatment ineligible patients—are not adequately explored.
-
There is no control group, and no analysis of downstream outcomes (e.g., resection rates, complications, or survival), which further weakens the impact of the findings.
To improve the manuscript, the authors should:
-
Clearly justify and/or exclude the non-smoking subgroup, or analyze it separately with appropriate caution.
-
Define more rigorously the inclusion/exclusion criteria.
-
Incorporate a broader literature base, especially from the European experience.
-
Expand the discussion on risks and limitations of screening in the elderly.
Reviewer 3 Report
Comments and Suggestions for Authors
This study investigates the effectiveness of low-dose CT (LDCT) screening for lung cancer among elderly male smokers aged 70 years and older. It provides clinically valuable insights into the detection rate and treatment effectiveness of lung cancer screening in an elderly population. Overall, the study is well-designed, and the results are clearly presented.
1. Redundancy in Statistical Analysis
The results section contains redundant descriptions of statistical significance, making it difficult to identify the most important findings. Emphasize the key statistically significant findings and present them concisely.
2. Unclear Structure in Discussion
Comparisons with major trials (NLST, NELSON) are mixed together, making the discussion hard to follow. Separate comparisons with each trial into distinct paragraphs, clearly highlighting the differences between this study and each of the major trials.
3. Lack of Clarity in Conclusions
Issue: The significance of LDCT screening for elderly individuals is not strongly emphasized. Highlight the importance of screening in elderly populations and the clinical implications of the study’s findings, while also suggesting future research directions.
4. Potential Selection Bias in Patient Recruitment
Issue: The study population consists of elderly male smokers who were able to visit a clinic for lung cancer screening, which may represent a relatively healthier subset. Explicitly state this selection bias and clarify that the findings may have limited generalizability to other populations.
5. Age Distribution Bias in Elderly Population
Issue: The study population is predominantly in the 70-74 age group, with very few participants aged 90 or older, limiting the generalizability of findings to the very elderly. Display the age distribution in a figure and consider further analysis of detection rates and treatment outcomes in the older age subgroups.
6. Short Follow-up Period for Detected Cancers
Issue: The median follow-up period is only 12 months, limiting the evaluation of long-term survival and treatment outcomes. Mention this limitation clearly in the discussion and suggest future studies with longer follow-up periods.
Round 2
Reviewer 1 Report
Comments and Suggestions for Authors
Please try to adress the issue in depth about the overdiagnosis issue. woud this issue be more significant or less in the old aged population. In addtion, no-smoking population data, no GGN lesion was reported in the non-smoking group, Please check data correctness. Otherwise the authors have revised the paper accroding to the reviewer comments. I have no additional comments.
Comments on the Quality of English Language
Please try to adress the issue in depth about the overdiagnosis issue. woud this issue be more significant or less in the old aged population. In addtion, no-smoking population data, no GGN lesion was reported in the non-smoking group, Please check data correctness. Otherwise the authors have revised the paper accroding to the reviewer comments. I have no additional comments.
Hsin-Hung , C., En-Kuei, T., Yun-Ju, W. et al. Impact of annual trend volume of low-dose computed tomography for lung cancer screening on overdiagnosis, overmanagement, and gender disparities. Cancer Imaging 24, 73 (2024). https://doi.org/10.1186/s40644-024-00716-5
Managing persistent subsolid nodules in lung cancer: education, decision making, and impact of interval growth patterns
YC Liu, CH Liang, YJ Wu, CS Chen, EK Tang, FZ Wu Diagnostics 13 (16), 2674
Reviewer 2 Report
Comments and Suggestions for Authors
We thank the authors for their efforts in revising the manuscript. However, the revised version still does not adequately address the major concerns previously raised. The study remains largely descriptive, with limited methodological depth and no significant advancement over existing literature or established clinical evidence. Additionally, key aspects of the design and analysis remain underexplored or insufficiently discussed. We encourage the authors to substantially reframe the work, emphasizing a clearer hypothesis, stronger methodology, and more in-depth discussion of the findings.
Round 3
Reviewer 2 Report
Comments and Suggestions for Authors
I appreciate the substantial effort you have made in revising the manuscript and in addressing the points raised during the previous rounds. I can see that you have clarified some methodological aspects and added details to support your approach.
However, I still have some lingering perplessities regarding the overall scientific message conveyed by the study. In particular, the inclusion of non-smokers in a screening cohort remains a point that I personally find difficult to fully endorse without a clearer epidemiological rationale. This choice, in my view, might introduce interpretative bias and reduce the applicability of the results to current screening recommendations.
Moreover, the strong emphasis placed on the potential benefits of a "one-shot" LDCT screening strategy seems, to me, somewhat optimistic considering the lack of robust survival data and the absence of a thorough analysis of potential harms and downstream implications.
The subgroup analyses you presented, while interesting, appear largely exploratory and may not be sufficiently powered to allow for definitive conclusions.
Overall, I believe that while your manuscript certainly addresses an important and timely topic, it would benefit from an even more cautious and balanced discussion to ensure that the scientific message aligns more clearly with the current evidence and clinical practice standards.
Author Response:
Response to Reviewer's Comments
We completely agree with the reviewer's comments and acknowledge that our current research, while offering valuable real-world data, has limitations in definitively demonstrating the efficacy of lung cancer screening in the elderly. We recognize that with this limited cohort data, our findings represent the best possible outcome for showcasing screening results within this specific population. Our intention was to provide detailed, focused data that would be beneficial to other researchers and inform future guidelines, even if the results are not yet definitive. The reviewer's guidance has been invaluable in helping us refine the manuscript's clarity and ensure its scientific message aligns more closely with current evidence and clinical practice.
We appreciate the reviewer's continued engagement with the overall scientific message and acknowledge the lingering perplexities, particularly regarding the inclusion of never-smokers and the interpretation of our "one-shot" LDCT screening strategy.
We have carefully revised the discussion section to address all specific points raised. We have:
Clarified the epidemiological rationale for including never-smokers: aiming to broaden the existing evidence base.
Adopted a more cautious and balanced tone regarding the "one-shot" LDCT strategy: We explicitly state that our study does not provide absolute evidence that lung cancer screening will yield benefits in this specific population. We also acknowledge that the absence of robust survival data is a key limitation necessitating careful interpretation of our findings.
Provided a more nuanced discussion of generalizability and potential biases: We now clearly state that our all-male, older study population limits generalizability to women and younger individuals. We also address the potential for selection bias, as participants likely represent a healthier and more health-conscious subset of the elderly, particularly those willing and able to attend a pulmonology clinic.
Highlighted the need for future research: We now explicitly call for future prospective studies with comprehensive data on both smokers and never-smokers, including long-term survival outcomes, to validate and expand upon our findings.
The revised paragraph in our discussion
"Our study contributes unique real-world evidence on the utility of LDCT screening in a demographically older and clinically distinct population, in contrast to previous trials such as NLST and NELSON, which excluded individuals aged ≥75 years or never-smokers. By including both elderly smokers and never-smokers, our study provides a more comprehensive view of screening performance across varying risk profiles. Meaningful detection rates and substantial treatment uptake—even among those aged ≥80 years—highlight the feasibility and clinical value of individualized screening strategies in routine practice.
However, caution is still warranted when applying our data to clinical practice, and several limitations should be carefully considered. This cohort, due to the inclusion of some never-smokers, cannot be considered fully representative of the entire smoking population, nor does it represent the full spectrum of never-smokers. Never-smokers were incorporated into our study design to generate much-needed screening data for this underrepresented population in current trials, but their number and the absence of survival rates necessitate careful interpretation of this study's findings. Furthermore, our study does not provide absolute evidence that lung cancer screening will yield benefits in this specific population. Importantly, the all-male, older study population limits generalizability to women and younger individuals. Additionally, participants were able to visit a clinic for screening, indicating they likely represent a healthier and more health-conscious subset of the elderly—particularly smokers willing and able to attend a pulmonology clinic—thus introducing potential selection bias. These factors should be meticulously considered when interpreting detection rates and applying our findings to broader lung cancer screening policies. Therefore, future prospective studies with comprehensive data on both smokers and never-smokers, including long-term survival outcomes, are essential to validate and expand upon our findings."
Please find the updated manuscript attached, with all newly added and revised sections highlighted in pink for your convenience.
We believe these revisions comprehensively address the valuable comments from the reviewer and yourself, presenting a more cautious, balanced, and nuanced discussion of our findings. We are confident that this strengthened discussion enhances the overall scientific message and applicability of our work.
Thank you again for your insightful feedback and the opportunity to improve our manuscript. We look forward to your further consideration.
Sincerely,